# Learning Repeatable Speech Embeddings Using An Intra-class Correlation Regularizer

**Jianwei Zhang**
School of Electrical, Computer and Energy Engineering
Arizona State University
Tempe, AZ 85281
jianwei.zhang@asu.edu

**Suren Jayasuriya**
School of Arts, Media and Engineering
School of Electrical, Computer and Energy Engineering
Arizona State University
Tempe, AZ 85281
sjayasur@asu.edu

**Visar Berisha**
College of Health Solutions
School of Electrical, Computer and Energy Engineering
Arizona State University
Tempe, AZ 85281
visar@asu.edu

## Abstract

A good supervised embedding for a specific machine learning task is only sensitive to changes in the label of interest and is invariant to other confounding factors. We leverage the concept of repeatability from measurement theory to describe this property and propose to use the intra-class correlation coefficient (ICC) to evaluate the repeatability of embeddings. We then propose a novel regularizer, the ICC regularizer, as a complementary component for contrastive losses to guide deep neural networks to produce embeddings with higher repeatability. We use simulated data to explain why the ICC regularizer works better on minimizing the intra-class variance than the contrastive loss alone. We implement the ICC regularizer and apply it to three speech tasks: speaker verification, voice style conversion, and a clinical application for detecting dysphonic voice. The experimental results demonstrate that adding an ICC regularizer can improve the repeatability of learned embeddings compared to only using the contrastive loss; further, these embeddings lead to improved performance in these downstream tasks.

## 1 Introduction

Embeddings, which are relatively low-dimensional latent representations of high-dimensional inputs, are widely used in deep learning applications and often trained through supervised learning techniques. In such cases, effective embeddings should be sensitive to changes in the target class (e.g., speaker identity, clinical class) while remaining invariant to unrelated factors (e.g., noise, natural variations of the data). Embeddings that satisfy this desired property have high *repeatability*, a term borrowed from measurement theory where it characterizes the consistency between the outcomes of consecutive

37th Conference on Neural Information Processing Systems (NeurIPS 2023).

measurements of the same target when the underlying conditions remain unaltered [23, 45]. For instance, in a text-independent speaker verification (TI-SV) task, the speaker embedding extracted from recordings with varying content from the same speaker should remain consistent. Similarly, in a dysphonic voice detection task, voice feature embeddings derived from the vowel phonation of a healthy person over consecutive days should remain consistent. By achieving high repeatability, embeddings effectively capture essential features while disregarding irrelevant factors.

A key to improving repeatability is reducing the intra-class variance, which is often increased by various confounding factors. Several studies have proposed different approaches to handling intra-class variation including increasing the variability of training data [60, 47, 7, 13, 61] and novel learning algorithms [2, 4, 42]. However, repeatability is rarely explicitly considered during training or evaluation of embeddings. In most cases, only the downstream applications' performance is used to indirectly evaluate the embeddings' effectiveness. We posit that directly assessing repeatability, regardless of downstream application, can help improve the quality of learned latent representations.

In this paper, we propose to use the intra-class correlation coefficient (ICC) to evaluate embeddings' repeatability. The ICC was designed to assess the consistency between two or more quantitative measurements [36], and often used to evaluate the repeatability of metrics across different fields [28, 33, 49]. We further propose a novel regularizer based on the ICC as a complementary component to traditional contrastive losses to enforce deep architectures to learn repeatable embeddings. Some contrastive losses, such as GE2E [54], push embeddings towards the centroid of the true class to reduce their intra-class variance. Via analysis and intuition, we explain why the ICC regularizer better focuses on minimizing the intra-class variance than contrastive loss alone and provide a new perspective for latent representation learning.

Repeatability is an especially difficult property to enforce in complex, high-dimensional signals like speech. Speech characteristics depend on the speaker's neurological and physiological state, the degrees of freedom in the speaking task, the recording setup, the environment, etc. [45]. These sources of variation challenge the development of embeddings for a particular task (e.g., learning speaker embeddings). We use the ICC regularizer to improve the embeddings' repeatability in three speech tasks: speaker embeddings for TI-SV, zero-shot voice style conversion, and voice feature embeddings for a clinical application of dysphonic voice detection. Our experimental results demonstrate that the proposed ICC regularizer can significantly improve the repeatability of learned embeddings, and embeddings with higher repeatability perform better in the downstream tasks. In the TI-SV task, the speaker embeddings' repeatability is significantly enhanced, and the EER decreases by $\sim 10\%$ compared to the methods without ICC regularizer. AB preference test results for zero-shot voice style conversion show that embeddings with higher repeatability are preferred 62% to 38% over those with lower repeatability. Objective evaluation metrics further confirm that more repeatable embeddings lead to improved performance in the voice style conversion task. In the clinical application, we demonstrate that highly repeatable voice feature embeddings further improve the in-corpus classification accuracy by $\sim 3\%$ and model generalizability across different corpora.

We summarize our contributions as follows:

- We connect the concept of *repeatability* from measurement theory to deep-learned embeddings and suggest the ICC for evaluating the quality of embeddings.

- We propose a novel regularizer, the *ICC regularizer*, as a complementary component for contrastive loss to regularize the deep-learned embeddings such that they are repeatable.

- We illustrate the reason why the ICC regularizer better minimizes the intra-class variance than the contrastive loss and provide a new perspective for latent representation learning.

- Our experimental results demonstrate that ICC regularizer can improve the repeatability of learned embeddings, and embeddings with higher repeatability exhibit better performance in downstream tasks.

For reproducibility of our work, the code for the ICC regularizer and experiments is available open-source in our GitHub repository[1].

---

[1] https://github.com/vigor-jzhang/icc-regularizer/

## 2 Related Work

**Learning invariant embeddings:** Previous literature has proposed to improve the robustness of learned latent representations by learning invariant embeddings. Most of these studies made the embeddings invariant to one or two types of variation, e.g. pose-invariance for re-identification [64, 35], noise-invariance for speaker recognition [7, 38], personality-invariance for emotion recognition [59]. There are two main approaches to promoting invariance to confounding factors: (1) increase the variability of training data, e.g., combining datasets from different domains [13], data warping [60, 47], data augmentation by GANs [61]; and (2) novel learning paradigms, such as variance-invariance-covariance regularization [4], invariant risk minimization [2], and simultaneously enforcing equivariance and invariance [42]. In this work, we follow the second approach by introducing a new regularizer for training embeddings.

**Intra-class and inter-class variance:** The variation between multiple observations of a class (intra-class variance) and the variation between classes (inter-class variance) define the performance of many machine learning criteria. Minimizing intra-class variance and maximizing inter-class variance are keys to many deep learning tasks, including classification [39, 63], representation learning [17, 29, 41], and few-shot learning [10, 8, 46]. However, these works typically rely on visualization (e.g., t-SNE, UMAP) to show how well their methods minimize intra-class variance and maximize inter-class variance [29, 34]. The proposed metric for repeatability, the ICC, can directly evaluate the performance of a given method using an already-established and well-understood metric from measurement theory. Furthermore, we can construct a regularizer centered around this metric.

**Contrastive loss:** Researchers have proposed finding task-relevant embedding features by contrastive representation learning and leveraging labeled data [50]. There are several popular contrastive losses in deep learning, such as the triplet loss [3, 31], tuple-based end-to-end (TE2E) loss [20], generalized end-to-end (GE2E) loss [54], momentum contrast (MoCo) [19], and SimCLR [9]. A contrastive loss encourages inputs of the same label class to have more similar latent representations compared to inputs from different classes. We select GE2E loss as a representative contrastive loss for comparison as it is widely adopted and used for supervised learning. The similarities and differences between the contrastive loss and the ICC regularizer are discussed in Section 3.2.

**Intra-class correlation coefficient:** Repeatability is most frequently measured via an intra-class correlation coefficient (ICC) [56]. Shrout and Fleiss elaborated several cases and corresponding formulas for ICC [43]. The ICC is used widely in different fields to estimate the reliability and repeatability, including clinical applications [45, 14], psychology and behavioral science [33, 15], and medical imaging [6, 49]. To the best of our knowledge, the ICC has not been applied for embeddings training and evaluation.

## 3 Method

In this section, we present the intra-class correlation coefficient (ICC) for evaluating the repeatability of deep-learned embeddings, and we present a novel ICC regularizer for enforcing repeatability in learned embeddings during training. Then, we illustrate the similarities and differences between contrastive loss and ICC regularizer by analyzing the intra-class and inter-class variance in relation to these two elements. Finally, we highlight why the ICC regularizer cannot function independently and its requirement for hyperparameter fine-tuning.

### 3.1 Intra-class correlation regularizer

Shrout and Fleiss elaborated several cases and corresponding formulas for the ICC [43], and we select the ICC 1-1 formulation, which is a measure of absolute agreement (i.e., the model generates the same embeddings to different samples from the same target) [16, 27], for assessing the repeatability of deep-learned embeddings.

In our problem formulation, we assume there are a set of high-dimensional data $\mathbf{x} \in \mathbb{R}^D$, each with label $\mathbf{y} \in [1, ..., N]$. The goal is to learn lower-dimensional latent representations $\mathbf{e} \in \mathbb{R}^L$ that exhibit maximum *separability*: embeddings belonging to the same class should be closely clustered together, while embeddings from different classes should be distinctly separated from each other [32]. An encoder with weights $\mathbf{w}$ and characterized by $f(\mathbf{x}; \mathbf{w})$, takes $\mathbf{x}$ as input and outputs

lower-dimensional representation $\mathbf{e}$. During training, the weights of the encoder are modified by optimizing a loss function that depends on $\mathbf{x}$ and labels $\mathbf{y}$. Considering a dataset with $\mathbf{y} \in [1, ..., N]$ and $M$ input samples per class, the embedding vector $\mathbf{e}_{ji}$ is defined as the $\ell_2$ normalization of the encoder output $f(\mathbf{x}_{ji}; \mathbf{w})$ $(1 \leq j \leq N, 1 \leq i \leq M, 1 \leq l \leq L)$:

$$\mathbf{e}_{ji} = [e_{ji}^1, e_{ji}^2, ..., e_{ji}^l, ...] = \frac{f(\mathbf{x}_{ji}; \mathbf{w})}{\|f(\mathbf{x}_{ji}; \mathbf{w})\|_2}, \tag{1}$$

where the $\mathbf{x}_{ji}$ represents the $i$-th sample of $j$-th class and the $\mathbf{e}_{ji}$ represents the corresponding embedding vector (the notation is similar to other contrastive losses), and $e_{ji}^l$ represents the $l$-th embedding dimension of $\mathbf{e}_{ji}$.

Then $ICC(e^l)$, the ICC score for $l$-th embedding dimension, can be calculated as follows [16]:

$$ICC(e^l) = \frac{MS_B(e^l) - MS_W(e^l)}{MS_B(e^l) + (M-1)MS_W(e^l)}. \tag{2}$$

Here the $MS_B(e^l)$ represents inter-class (between-class) variance[2] and $MS_W(e^l)$ represents the intra-class (within-class) variance for $l$-th embedding dimension. $MS_B(e^l)$ is calculated as

$$MS_B(e^l) = \frac{M \cdot \sum_{j=1}^N (\overline{e_j^l} - \overline{e^l})^2}{N-1}, \tag{3}$$

where $\overline{e_j^l} = \sum_i^M e_{ji}^l / M$ represents the mean of $l$-th embedding dimension for the $j$-th class and $\overline{e^l} = \sum_j^N \sum_i^M e_{ji}^l / (N \times M)$ represents the overall mean of $l$-th embedding dimension. Intuitively, $MS_B(e^l)$ measures the inter-class variance of $l$-th embedding dimension.

$MS_W(e^l)$ is calculated as

$$MS_W(e^l) = \frac{\sum_j^N M\sigma_{j,l}^2}{N(M-1)}, \tag{4}$$

where $\sigma_{j,l}^2 = \sum_i^M (e_{ji}^l - \overline{e_j^l})^2 / M$ represents the variance of $l$-th embedding dimension for $j$-th class. Intuitively, $MS_W(e^l)$ measures the overall intra-class variance of $l$-th embedding dimension.

After obtaining the ICC score for each embedding point, we use the mean value of all embedding dimensions' ICC scores as the repeatability metric:

$$ICC(\mathbf{e}) = \frac{\sum_l^L ICC(e^l)}{L}. \tag{5}$$

**ICC score interpretation**: When the learned embeddings exhibit perfect repeatability (characterized by intra-class variance $MS_W = 0$ and inter-class variance $MS_B > 0$), the ICC score is equal to 1. A decrease in the ICC score signifies reduced repeatability, which implies a relative increase in intra-class variance $MS_W$ compared to inter-class variance $MS_B$. If the intra-class variance $MS_W$ exceeds the inter-class variance $MS_B$, the ICC score may become negative[3].

We propose a novel regularizer, the ICC regularizer, for regularizing learned representations to enforce high repeatability. The ICC regularizer operates on each batch: we assume a single batch contains samples from $N$ classes, and $M$ input samples from each class, and the dimension of the embedding is $L$. The ICC loss firstly uses Equation 5 to calculate the mean ICC score $ICC(\mathbf{e})$ for the embeddings $\mathbf{e}$ of current batch, and then the ICC regularizer can be written as

---

[2]Often noted as $MS_R$ in ICC related literature [43, 16].

[3]For instance, consider a simple numerical example where there is $N = 2$ classes and $M = 6$ samples per class. Class 1 has mean $\mu_1 = 0$ and variance $\sigma_1^2 = 100$, class 2 has mean $\mu_2 = 0.1$ and variance $\sigma_2^2 = 100$. In this situation, $MS_B = 0.3$, $MS_W = 120$, then $ICC = -0.199 < 0$. $M$ has big effect in this situation: when $M$ is large, the ICC score is still negative but very close to 0.

$$R_{ICC} = 1 - ICC(\mathbf{e}). \tag{6}$$

In this paper, the ICC and ICC regularizer require equal class size, i.e., $M$ is the same for all classes. However, to expand the ICC regularizer usage for scenarios where the number of samples per class may not be equal, we provide an extended version of the ICC formulation and code for the imbalanced classes in the Appendix A and in our GitHub repository[4].

## 3.2 ICC regularizer vs. contrastive loss

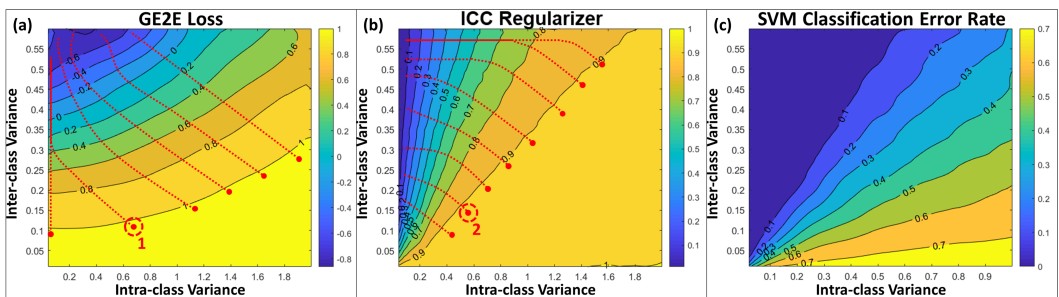

Figure 1: The contour figures for (a) GE2E loss and (b) ICC regularizer value of intra- and inter-class variance. To explore the gradient trend of GE2E loss and ICC regularizer, some starting points (red dots) are selected, then the maximum gradient descent path (red dashed lines) of value is calculated and plotted. (c) The contour figure for SVM classification error rate on simulation data per intra- and inter-class variance.

The ICC regularizer and contrastive loss have similarities in their optimization criteria: both aim to minimize the intra-class variance and maximize the inter-class variance. However, they exhibit different tradeoffs between the two variances. We use a Monte Carlo simulation to study the similarities and differences of the ICC regularizer and contrastive loss on the intra-class and inter-class variance. We use the GE2E loss [54] as a representative contrastive loss in simulation[5].

**Simulation setup:** In simulation, we vary the intra- and inter-class variances to generate samples as follows: (1) the intra-class variance, i.e., the variance of simulated embeddings within one class, varies from 0.02 to 2.0 with a step size of 0.02; (2) the inter-class variance, i.e., the variance of class centroids, varies from 0.01 to 0.60 with a step size of 0.01. For a pair of configurations (e.g. inter-class, intra-class pair), we draw 400 samples from an 8-dimensional, 4-class Gaussian mixture such that the class-conditional mean and variance yield the desired inter-class, intra-class variance pair. We calculate the ICC regularizer value and GE2E loss value for these random samples. We repeat the Monte Carlo simulation 100 times and plot the loss values in Figure 1 for the ICC regularizer and GE2E loss as a function of the inter-class and intra-class variance. To explore their landscapes, we select several starting points, then trace the path of maximum gradient descent.

**Discussion:** The ICC regularizer and GE2E loss have very different contours. They both have a low value when the inter-class variance is high and the intra-class variance is low. However, when embeddings are normalized (see Equation 1), the normalization places an upper limit on the total variance of embeddings and limits how large the inter-class variance can be (the inter-class variance cannot go to infinity). So, a regularizer that places a greater emphasis on minimizing the intra-class variance for a bounded inter-class variance will naturally lead to embeddings with higher repeatability compared to a loss that focuses on simultaneously maximizing both variances.

The GE2E loss tries to optimize both intra-class and inter-class variances simultaneously: the maximum gradient descent direction of the GE2E loss overall is from the lower-right corner to the upper-left corner. This loss continues to decrease as the inter-class variance increases, even after the intra-class variance is small enough (the descent path of point 1 in Figure 1(a)). We note that the

---

[4]https://github.com/vigor-jzhang/icc-regularizer/

[5]We implement the GE2E loss with 'softmax' type, i.e., using Equation 6 in their paper. The GE2E loss use two learnable parameters $w, b$ for their similarity metric calculation $S = w \cdot cos(\cdot) + b$, but we fixed $w = 10, b = -5$ as initial values suggested in their paper for simulation.

embeddings of different classes are already clustered well under this scenario. Therefore, further increasing the inter-class variance does not improve the separability between the embeddings as they are already separated. It is difficult for the GE2E loss to further minimize the intra-class variance without increasing the inter-class variance based on the contours of the loss. Our simulation explains findings from several studies showing that contrastive losses, including the GE2E, do not perform well in reducing intra-class variance [30, 55].

In contrast, the ICC regularizer focuses on decreasing the intra-class variance and places less emphasis on the inter-class variance, thereby enhancing repeatability of the learned embeddings. The minimum for the ICC regularizer occurs when the embeddings' intra-class variance is approximately equal to 0, or the inter-class variance reaches a relatively large value compared to the intra-class variance. The ICC regularizer pushes the embeddings towards lower intra-class variance and not towards larger inter-class variance once the inter-class variance exceeds the intra-class variance (ensuring the embeddings are clustered well) as shown in Figure 1 (b). This naturally leads to embeddings with better repeatability compared to the GE2E loss.

A simple analysis of gradients explains this observation. The gradient of the ICC regularizer with respect to the two variance terms is

$$\frac{\partial R_{ICC}}{\partial MS_B} = -\frac{M \times MS_W}{(MS_B + (M-1) \times MS_W)^2}, \tag{7}$$

$$\frac{\partial R_{ICC}}{\partial MS_W} = \frac{M \times MS_B}{(MS_B + (M-1) \times MS_W)^2}. \tag{8}$$

When the intra-class variance ($MS_W$) is already small, the derivative of the ICC regularizer with respect to the inter-class variance ($MS_B$) is small in absolute value; therefore, the gradient descent step along the inter-class variance dimension is small. When the inter-class variance is relatively large, the derivative of the ICC regularizer with respect to the intra-class variance is larger, which means the gradient descent step along the intra-class variance dimension is relatively large. This is clear from Figure 1 (b) where the points that begin with a small intra-class variance (e.g. point 2 in Figure 1 (b)) do not focus on further increasing the inter-class variance to improve the ICC as it is unnecessary. In contrast, the trajectory of point 1 in 1 (a) goes towards increasing the inter-class variance, even when the intra-class variance is small and the clusters are already well separated.

The ICC regularizer is also better aligned with classification error rate. We use the same simulation data and train a SVM for each intra- and inter-class variance pair to generate the contour figure for SVM classification error rate. As we show in Figure 1 (c), the classification error rate has a similar contour when compared with the ICC regularizer values: the lower ICC regularizer (higher the repeatability), the lower the classification error rate. This result supports our hypothesis that embeddings with improved repeatability will benefit downstream applications.

In summary, given the constraints on the total variance of the embeddings, it is beneficial to focus on the intra-class variance for increasing representation repeatability. Although the optimization objectives of contrastive loss and ICC regularizer are similar, the contrastive loss focuses on optimizing intra-class and inter-class variance simultaneously whereas the ICC regularizer places greater emphasis on minimizing intra-class variance.

**The ICC is not a replacement for contrastive loss**: The ICC is computed for each dimension of the embedding independently, while contrastive loss is calculated for the entire embedding vector. It's important to note that high separability in each embedding dimension doesn't necessarily translate into good separability in the overall embedding space and can lead to learned embedding dimensions that are highly correlated; this can have a negative impact on downstream model performance [51]. Relatedly, the cosine similarity score is commonly used to evaluated the fidelity of learned embeddings. A high ICC value per dimension does not necessarily imply a high cosine similarity. For these reasons, we propose to use the ICC as a regularizer rather than a stand-alone loss.

**The ICC regularizer requires hyperparameter fine-tuning**: As with other regularizers, a linear combination of contrastive loss and the ICC regularizer, represented by the equation $L_{contr} + \lambda R_{ICC}$, requires selection of the hyperparameter $\lambda$. In our experiments, we used a grid search to determine the optimal $\lambda$. For visualizations of the loss contour (similar to those in Figure 1) of the combined

contrastive+ICC loss, we refer the reader to the Appendix B where we have included the combined loss contour with varying $\lambda$ values.

## 4 Experimental Results

In this section, we regularize deep-learning models with the ICC to improve the embeddings' repeatability in three different speech tasks: (1) speaker embeddings for text-independent speaker verification, where the ICC regularizer ensures the embeddings are repeatable for the same person and the contrastive loss aims to separate embeddings of different speakers; (2) speaker embeddings for zero-shot voice style conversion, where using embeddings with better quality results in higher-quality conversion; (3) voice feature embeddings for dysphonic voice detection, where the ICC regularizer ensures the embeddings do not change from day to day for the healthy group and a contrastive loss forces maximum separability between dysphonic and healthy speech.

### 4.1 Task 1: text-independent speaker verification

Text-independent speaker verification (TI-SV) systems verify the speaker's identification using a speech signal input without any constraints on the speech content. Many previous methods use speaker embeddings for the TI-SV task [54, 24, 48, 26]. For TI-SV, the speaker embeddings should only capture the difference in the speaker identities and be invariant to all other confounding factors, including utterance content and length. Herein we include our proposed ICC regularizer when learning speaker embeddings to improve their repeatability.

**Experiment setup:** We select three well-known contrastive losses as baselines: (1) GE2E [54], (2) Angular Prototypical (AngleProto) [12], and (3) SupCon [25]. Then we use these contrastive losses with and without the ICC regularizer to train two encoders, VGG-M-40 and FastResNet-34 which is described in Chung et al. paper [12], and compare the performance and quality of these speaker embeddings. We use the VoxCeleb 1 & 2 development dataset for training, and VoxCeleb 1 testing dataset for TI-SV performance evaluation [37, 11]. For training VGG-M-40, each batch contains $N = 8$ speakers and $M = 30$ utterances per speaker, and the loss formula $L(\mathbf{e}) = 1.0 \times L_{contr}(\mathbf{e}) + 0.06 \times R_{ICC}(\mathbf{e})$, where the $\mathbf{e}$ is the embeddings of speakers' in one batch. For training FastResNet-34, each batch contains $N = 100$ speakers and $M = 2$ utterances per speaker, and the loss formula $L(\mathbf{e}) = 1.0 \times L_{contr}(\mathbf{e}) + 0.25 \times R_{ICC}(\mathbf{e})$. During training, we use the Adam optimizer, maintaining a static learning rate of 0.001 without implementing any learning rate schedule. The dropout rate is set to 0.2 for all dropout layers. As for data augmentation: (1) we use variation in input audio length by randomly fixing the audio duration within a range of 1.5 to 3.0 seconds, and (2) we add Gaussian noise with a SNR randomly selected between 15 to 60 dB. No other augmentation methods are used. The hyper-parameter is tuned on the development dataset. The EER for subjects in the development dataset are used to determine the optimized hyperparamter. We use the EER and minDCF to evaluate the performance of TI-SV, ICC (Equation 5) to evaluate the repeatability of embeddings.

Table 1: TI-SV task EER and ICC results for contrastive losses with and without ICC regularizer.

| | VGG-M-40 | | | FastResNet-34 | | |
|---|---|---|---|---|---|---|
| | EER | minDCF | ICC | EER | minDCF | ICC |
| GE2E [54] | 4.39% | 0.2925 | 0.4494 | 2.49% | 0.2133 | 0.7215 |
| GE2E + ICC | 3.96% | 0.2778 | 0.5487 | 2.39% | 0.2012 | 0.7366 |
| AngleProto [12] | 4.36% | 0.2809 | 0.4399 | 2.28% | 0.1960 | 0.7501 |
| AngleProto + ICC | 4.02% | 0.2790 | 0.5455 | 2.17% | 0.1871 | 0.7627 |
| SupCon [25] | 3.91% | 0.2791 | 0.5693 | 2.30% | 0.1956 | 0.7500 |
| SupCon + ICC | **3.78%** | **0.2597** | **0.6661** | **2.16%** | **0.1867** | **0.7615** |

**Results:** Table 1 presents the results of the TI-SV evaluation for contrastive losses, both with and without the ICC regularizer. When using VGG-M-40 and FastResNet-34 models, the GE2E achieves EER of 4.39% and 2.49% on the VoxCeleb 1 test set, respectively. Incorporating the

ICC regularizer improves the GE2E's performance, reducing EER to 3.96% for VGG-M-40 and 2.39% for FastResNet-34. This represents a approximately 10% enhancement in performance for the VGG-M-40 model. The repeatability of speaker embeddings also improves with the ICC regularizer as the ICC score is increased from 0.4494 to 0.5487 and from 0.7215 to 0.7366 for VGG-M-40 and FastResNet-34, respectively. In addition, the benefits of the ICC regularizer are also observed with two other contrastive loss methods. For the AngleProto method, introducing the ICC regularization achieves a 7.8% and 4.8% improvement over baseline models without ICC regularizer. Meanwhile, the SupCon method, records a 3.3% and 6.08% improvement over SupCon without ICC regularizer, also resulting in more repeatable embeddings.

The experimental results of the TI-SV task demonstrate that the proposed ICC regularizer can improve the repeatability of embeddings learned to be sensitive to speaker identities. The speaker embeddings with higher repeatability achieve improved performance on the TI-SV task.

### 4.2 Task 2: zero-shot voice style conversion

One important application of speaker embeddings is voice style conversion, i.e., modifying the speech of a source speaker to sound like that produced by another target speaker without changing the linguistic information [40, 21, 58]. In the previous section, we used the ICC regularizer together with a contrastive loss during training to obtain speaker embeddings with higher repeatability. The embeddings with high repeatability should benefit downstream applications. While in the previous section we demonstrated that they benefit the TI-SV task for several different contrastive losses, in this section, we use a zero-shot voice conversion model, AutoVC [40], and evaluate it with two different speaker embeddings generated from the previous section: (1) GE2E loss trained speaker embeddings which are less repeatable, and (2) GE2E + ICC regularizer trained speaker embeddings which are more repeatable. We choose the most challenging voice style conversion task, the zero-shot voice style conversion (unseen speaker to unseen speaker), to compare the downstream performance of these two embeddings with different repeatability.

**Experiment setup:** We use exactly same training procedure described in Qian et al. [40] to train the two models. For evaluating the conversion quality perspectively, we conduct AB preference test to compare the generated samples: we randomly select 10 unseen source and 10 unseen target speakers from the VCTK corpus [52], to generate a total of 100 source-target speaker pairs[6]. For each listening test, 15 pairs are randomly selected from the 100 pairs. The order of presentation is randomized. A total of 18 listeners participated in this AB preference test. They were instructed to select the sample that better matched the target speaker without knowing what method was used to generate the samples. We also evaluate two methods objectively by using objective scores based on the word error rate (WER) and character error rate (CER). We use an opensource speaker encoder[7] to calculate the speaker similarity score between target speaker's audio and transformed output. And we then use Wav2Vec2[8] to do ASR on the transformed output, and calculate the WER and CER by using *jiwer* module[9].

Table 2: AB preference result for zero-shot voice style conversion.

|  | GE2E Loss | **GE2E Loss + ICC Regularizer** |
|---|---|---|
| Selection Ratio | 38.10% | **61.90%** |

Table 3: Objective evaluation result for zero-shot voice style conversion.

|  | **Speaker Similarity Score** | **WER** | **CER** |
|---|---|---|---|
| **GE2E Loss** | 0.2231 | 0.5810 | 0.3817 |
| **GE2E Loss + ICC** | 0.2309 | 0.5109 | 0.3324 |

**Results:** The AB preference result is shown in Table 2. On average, 61.90% of samples generated by the model trained with the more repeatable speaker embeddings were preferred over those trained

---

[6]We provide several examples on our project page: https://vigor-jzhang.github.io/icc-reg-project-page/.

[7]huggingface.co/speechbrain/spkrec-ecapa-voxceleb

[8]huggingface.co/docs/transformers/model_doc/wav2vec2

[9]pypi.org/project/jiwer/2.5.1/

using the original GE2E loss speaker embeddings. The objective evaluation results are shown in Table 3. The objective evaluation metrics demonstrate that speaker embeddings with higher repeatability also result in better performance across all these metrics for voice style conversion. All these results demonstrate that speaker embeddings with higher repeatability also result in better performance for voice style conversion task.

### 4.3 Task 3: assessment of vocal quality for dysphonic voice detection

Dysphonia is a term that refers to difficulty producing clear voicing during speech production. Automatic dysphonic voice detection by deep learning has attracted academic and clinical interest. To develop reliable clinical models, it is important to use highly repeatable voice features and embeddings [45]. Zhang et al. proposed voice feature embeddings sensitive to vocal quality and robust across different corpora [62]. However, in their work there is no constraint on the voice feature embeddings' repeatability.

We rebuild the Zhang et al. network and implement the ICC regularizer to enhance the repeatability of embeddings. We follow the procedure described in [62] for training the voice feature embeddings. To enforce repeatability using the ICC regularizer, we use the recordings of healthy subjects from the mPower corpus [5] to ensure the embeddings do not change daily for the healthy person.

**Model structure:** We use the same encoder and MLP classifier from the Zhang et al. paper. A two-branch structure is used for optimizing the dysphonic sensitivity and repeatability of voice feature embeddings simultaneously. All three embeddings have a dimension of 256. For more information about our model structures, we refer the reader to the Appendix C.

**Training loss:** We use the following formula for training: $L = 0.5 \times R_{ICC} + 1.0 \times L_{contr} + 1.0 \times L_{class}$, where the $R_{ICC}$ is the ICC regularizer on repeat-constrained embeddings, $L_{contr}$ is the contrastive loss on the voice feature embeddings, and $L_{class}$ is the classification loss.

**Training and evaluation datasets:** We use the Saarbruecken Voice Database (SVD) [57] as the training and in-corpus validation dataset for dysphonic voice detection, and the mPower corpus [5] is used only for improving the repeatability of voice feature embeddings. The Massachusetts Eye and Ear Infirmary (MEEI) database and Hospital Príncipe de Asturias (HUPA) [1] dataset are used for cross-corpus testing datasets for dysphonic voice detection task, and the ALS [44] dataset is used for repeatability evaluation. The MEEI, HUPA, and ALS are unseen during training for all methods. A summary of these datasets are provided in Appendix D.

**Training details:** We perform cross-validation six times to characterize the variability in performance. We fixed the random seed to 233. The SGD optimizer is used with a learning rate of 0.001 and other default settings. We use one NVIDIA Titan Xp graphic card to train our models. We train the model for 20k steps, which takes approximately 16 hours under our configurations.

**Baseline methods:** We compare against four baselines: (1) Zhang et al. [62]; (2) P. Harar et al. [18], which is based on a recurrent convolutional neural network model; (3) L. Verde et al. [53], which used a conventional features set with different classical machine learning classifiers; (4) M. Huckvale et al. [22], which uses the ComPare feature set from the OpenSMILE toolkit with the SVM and neural networks methods. All baseline methods are rebuilt and trained on our data using the same procedures as they outlined in the original papers.

**Evaluation metrics:** We use balanced accuracy to evaluate the dysphonic voice classification accuracy. We use the ICC (Equation 5) to evaluate the repeatability of our trained voice feature embeddings and other baseline methods' features. For a fair comparison, we evaluate the repeatability by using ALS dataset [44], which is unseen to all methods during training.

Table 4: Dysphonic voice detection accuracy and features' repeatability of our and baseline methods.

| | Dysphonic Voice Classification / [mean accuracy] (95% CI) | | | | Repeatability / ICC |
| | SVD Train | SVD Validation | MEEI Testing | HUPA Testing | ALS |
|---|---|---|---|---|---|
| **Proposed Method** | 0.7353 (0.004) | **0.7289 (0.009)** | **0.8214 (0.004)** | **0.6894 (0.011)** | **0.5708** |
| Zhang et al. (2022) | 0.8003 (0.018) | 0.7077 (0.011) | 0.8209 (0.014) | 0.6651 (0.008) | 0.4368 |
| Harar et al. (2017) | 0.7742 (0.017) | 0.6914 (0.009) | 0.6614 (0.024) | 0.4918 (0.008) | 0.4743 |
| Verde et al. (2018) | 0.8910 (0.006) | 0.6274 (0.009) | 0.7042 (0.018) | 0.5976 (0.015) | 0.0182 |
| Huckvale et al. (2021) | 0.7290 (0.019) | 0.6255 (0.012) | 0.6978 (0.044) | 0.5487 (0.014) | 0.2914 |

**Results:** Implementing the ICC regularizer significantly improves the repeatability of the voice feature embeddings as shown in Table 4. Our proposed method achieves the highest ICC score of 0.5708, while the Zhang et al. model only achieves an ICC score of 0.4368, an improvement of 30.68%. Similarly, the repeatability of embeddings generated with the ICC regularizer regularizer significantly exceeds that of all baseline methods.

Our experimental results demonstrate that the voice feature embeddings with higher repeatability also achieve better classification accuracy and generalizability. Our proposed method achieves good classification accuracy on SVD in-corpus validation 0.7289 ($\pm$0.009), MEEI cross-corpus testing 0.8214 ($\pm$0.004), and HUPA cross-corpus testing 0.6894 ($\pm$0.011). For a comparison, the original HUPA publication achieved accuracy of 0.6962 ($\pm$0.047) with MFCC when trained and tested in-corpus [1]. Our proposed method's classification accuracy across the three corpora is quite good, considering the differences between the corpora.

The experimental results of voice feature embeddings for dysphonic voice detection task demonstrate that the ICC regularizer can improve the repeatability of embeddings and embeddings with higher repeatability exhibit better accuracy and generalizability in dysphonic voice detection.

## 5   Conclusion

This paper ports the concept of repeatability from measurement theory to representation learning. We propose to use the ICC as an evaluation metric in representation learning and use the ICC regularizer as a complementary component for contrastive loss to regularize deep-learned embeddings to be more repeatable. We use an example and intuition to explain why the ICC regularizer has better performance on minimizing intra-class variance than contrastive loss. We evaluate the ICC regularizer on three speech tasks that use learned embeddings: speaker embeddings for TI-SV and zero-shot voice style conversion, and voice feature embeddings for a clinical application. The experimental results demonstrate that the ICC regularizer can improve the repeatability of learned embeddings, and embeddings with higher repeatability exhibit better performance in downstream tasks. There several directions for future works: (1) applying the ICC regularizer to other domains, including computer vision and natural language processing; (2) extension to self-supervised methods that use contrastive-style training; (3) a more thorough theoretical analysis of the ICC and its properties.

**Potential Negative Societal Impact:** Methods for learning new feature representations that focus on separability between classes can amplify biases that exist in the data. This is a well-known problem and it can occur when the data used to train the representation model is biased. This is especially problematic in high-stakes applications like healthcare, where biased predictions or decisions can lead to unequal treatment or access. Safe deployment of models based on the feature representations proposed herein will require thorough validation to detect potential biases and mitigation strategies for dealing with them.

## Acknowledgments and Disclosure of Funding

This work was funded in part by Office of Naval Research grants N00014-21-1-2615 and N00014-23-1-2406. The authors acknowledge Research Computing at Arizona State University for providing GPU resources that have contributed to the research results reported within this paper.

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

# Appendix

## A    ICC Formulation for Imbalanced Classes

The proposed ICC and ICC regularizer in the main paper require equal class size, i.e., $M$ is the same for all classes. However, this is not typically the case in actual usage. In this section, we propose an extended version of the ICC and ICC regularizer, which are capable of handling datasets with unbalanced class sizes.

Assume there are $N$ classes in the dataset or batch, and $k_j$ is the number of samples for $j$-th class, the $\mathbf{x}_{ji}$ represents the $i$-th samples of the $j$-th class, $\mathbf{e}_{ji}$ represents the corresponding embedding vector, and $e_{ji}^l$ represents the $l$-th embedding dimension of the $\mathbf{e}_{ji}$. The between-class variance can be written as:

$$MS_B(e^l) = \frac{\sum_{j=1}^{N} k_j \cdot (\overline{e_j^l} - \overline{e^l})}{N-1}, \tag{9}$$

where $\overline{e_j^l}$ represents the mean of $l$-th embedding dimension for the $j$-th class:

$$\overline{e_j^l} = \frac{\sum_{i=1}^{k_j} e_{ji}^l}{k_j}, \tag{10}$$

and $\overline{e^l}$ represents the overall mean of $l$-th embedding dimension,

$$\overline{e^l} = \frac{1}{N} \sum_{j=1}^{N} \frac{\sum_{i}^{k_j} e_{ji}^l}{k_j}. \tag{11}$$

Then $ICC(e^l)$, the ICC score for $l$-th embedding dimension, can be calculated as follows:

$$ICC(e^l) = \frac{MS_B(e^l) - \frac{1}{N} \sum_{j=1}^{N} \frac{\sum_{i=1}^{k_j} \sigma_{ji,l}^2}{k_j-1}}{MS_B(e^l) + \frac{1}{N} \sum_{j=1}^{N} \sum_{i=1}^{k_j} \sigma_{ji,l}^2}, \tag{12}$$

where $\sigma_{ji,l}^2 = (e_{ji}^l - \overline{e_j^l})^2$ represents the within-class variance of $l$-th embedding dimension for $i$-th sample of the $j$-th class. The Equation 12 computes the intra-class variance for each class individually, taking into account the number of samples present in each class, denoted as $k_j$. Thus, Equation 12 effectively addresses issues related to class size imbalance.

Then the ICC score and ICC regularizer still can be calculated by using

$$ICC(\mathbf{e}) = \frac{\sum_{l}^{L} ICC(e^l)}{L}, \tag{13}$$

$$R_{ICC} = 1 - ICC(\mathbf{e}), \tag{14}$$

respectively, for the datasets with unbalanced class sizes.

## B    Hyperparamter Ablation for GE2E Loss with ICC Regularization

As described in the Section 3.2 of the main paper, the ICC regularizer cannot be used alone and requires hyperparameter fine-tuning when combined with the contrastive loss. Therefore, based on the simulation data in Section 3.2 in the paper, the simulation contour figures of the $(1-\lambda)L_{GE2E} + \lambda R_{ICC}$ function value of intra- and inter-class variance for different values hyperparameter $\lambda$ are provided in Figure B.1.

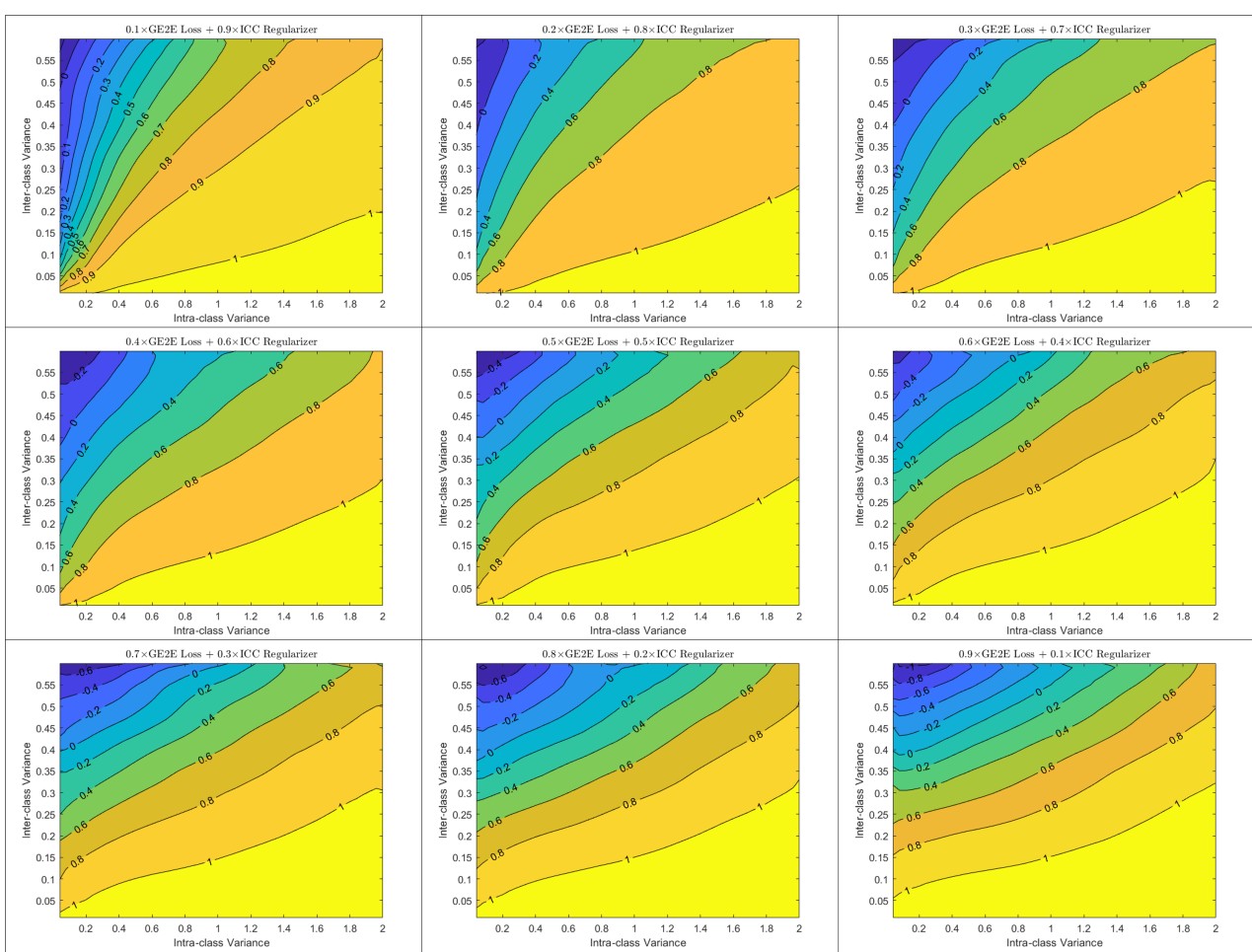

Figure B.1: The simulation contour figures of the $(1 - \lambda)L_{GE2E} + \lambda R_{ICC}$ function value of intra- and inter-class variance, where $\lambda = \{0.1, 0.2, 0.3, 0.4, 0.5, 0.6, 0.7, 0.8, 0.9\}$.

Referring to Figure B.1, it's observed that the impact of the hyperparameter $\lambda$ on the shape of the $(1 - \lambda)L_{GE2E} + \lambda R_{ICC}$ function's contour lines is linear under the given simulation conditions. As the value of this hyperparameter $\lambda$ increases, the contour of $(1 - \lambda)L_{GE2E} + \lambda R_{ICC}$ leans more towards the ICC regularizer. This means it increasingly emphasizes on reducing the intra-class variance.

## C  Model and Training Details for Task 3: Dysphonic Voice Detection

We rebuild the Zhang et al. network [62] and implement the ICC regularizer to enhance the repeatability of voice feature embeddings. We follow the procedure described in their work [62] for training the voice feature embeddings. To enforce repeatability using the ICC regularizer, we use the recordings of healthy subjects from the mPower corpus [5] to ensure the embeddings do not change daily for the healthy person.

**Model Structure:** We use the same encoder and MLP classifier from the Zhang et al. paper [62]. A two-branch structure is used for optimizing the dysphonic sensitivity and repeatability of voice feature embeddings simultaneously as shown in Figure C.2, where the MLP networks is composed by three linear layers and two Leaky-ReLU activation layers (negative slope is 0.4). All three embeddings have a dimension of 256.

**Training Loss:** We use the following loss:

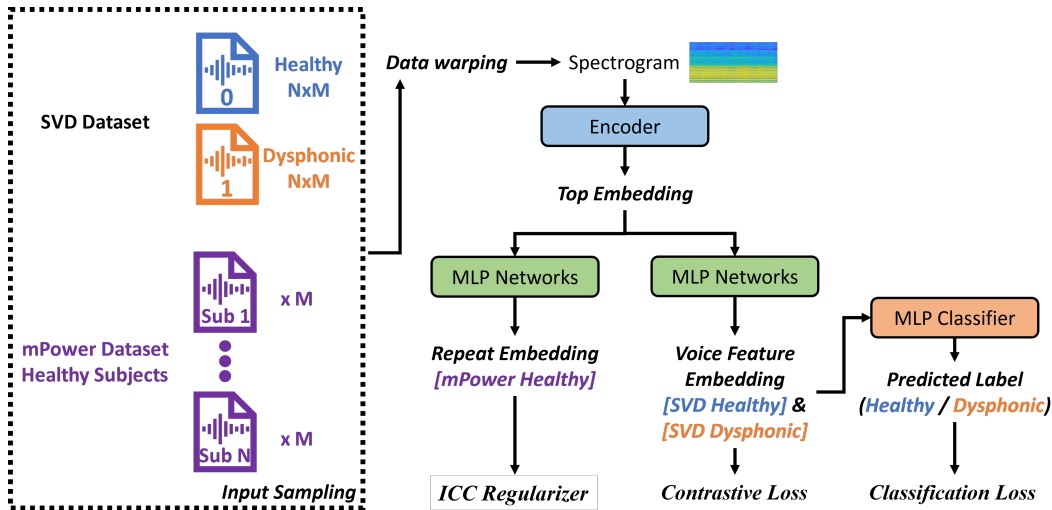

Figure C.2: The diagram of training repeatability enhanced voice feature embeddings for dysphonic voice detection.

$$L = 0.5 \times R_{ICC} + 1.0 \times L_{contr} + 1.0 \times L_{class}, \tag{15}$$

where the $R_{ICC}$ is the ICC regularizer on repeat-constrained embeddings, $L_{contr}$ is the contrastive loss on the voice feature embeddings, and $L_{class}$ is the classification loss.

**Training and Evaluation Datasets:** We use the Saarbruecken Voice Database (SVD) [57] as the training and in-corpus validation dataset for dysphonic voice detection task, and the mPower corpus [5] is used only in training for improving the repeatability of voice feature embeddings. The Massachusetts Eye and Ear Infirmary (MEEI) database and Hospital Príncipe de Asturias (HUPA) [1] dataset are used for cross-corpus testing datasets for dysphonic voice detection task, and the ALS [44] dataset is used for repeatability evaluation. The MEEI, HUPA, and ALS are unseen during training for all methods.

**Training Details:** We perform cross-validation six times to characterize the variability in performance. We fixed the random seed to 233. For each training batch, we randomly select 16 dysphonic and 16 healthy voice recordings of the same gender from the SVD dataset; 8 healthy subjects from the mPower dataset, and 2 consecutive days' voice recordings for each subject. The SGD optimizer is used with a learning rate of 0.001 and other default settings. We use one NVIDIA Titan Xp graphic card to train our models. We train the model for 20k steps, which takes approximately 16 hours under our configurations.

**Baseline Methods:** We comapre against four baselines: (1) Zhang et al. [62]; (2) Harar et al. [18], which is based on a recurrent convolutional neural network model; (3) Verde et al. [53], which used a conventional features set with different classical machine learning classifiers; (4) Huckvale et al. [22], which uses the ComPare feature set from the OpenSMILE toolkit with the SVM and neural networks methods. All baseline methods are rebuilt and trained on our data using the same procedures as they outlined.

**Evaluation Metrics:** We use balanced accuracy to evaluate the dysphonic voice classification accuracy:

$$\text{Balanced accuracy} = \frac{1}{2} \times (\frac{\text{TP}}{\text{TP} + \text{FN}} + \frac{\text{TN}}{\text{TN} + \text{FP}}). \tag{16}$$

We use the ICC to evaluate the repeatability of our trained voice feature embeddings and other baseline methods' features. For a fair comparison, we evaluate the repeatability by using ALS dataset, which is unseen to all methods during training.

## C.1 Why use a two-branch structure?

We use a two-branch structure for optimizing the dysphonic sensitivity and repeatability of embeddings simultaneously: the top embeddings generated by the encoder are further transformed to repeat-constrained embeddings and voice feature embeddings instead of directly using top embeddings for repeatability constraints and dysphonic voice detection.

We have tested three models with different structures in this experiment, including: (1) one-embedding output, i.e., Zhang et al. networks structure without modification and there is only one embeddings output can be optimized as shown in Figure C.3; (2) two-embedding output, i.e., add one MLP network to transform the top embeddings to the voice feature embeddings and there are two embeddings outputs can be optimized as shown in Figure C.4; (3) a two-branch structure (three-embedding output).

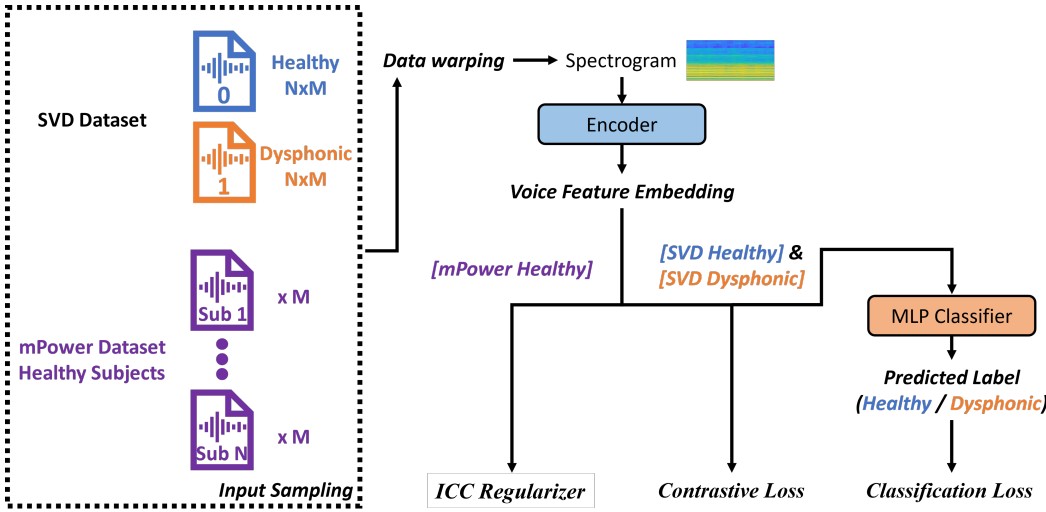

Figure C.3: The diagram of one-embedding output model structure. This is not used for any results in this paper.

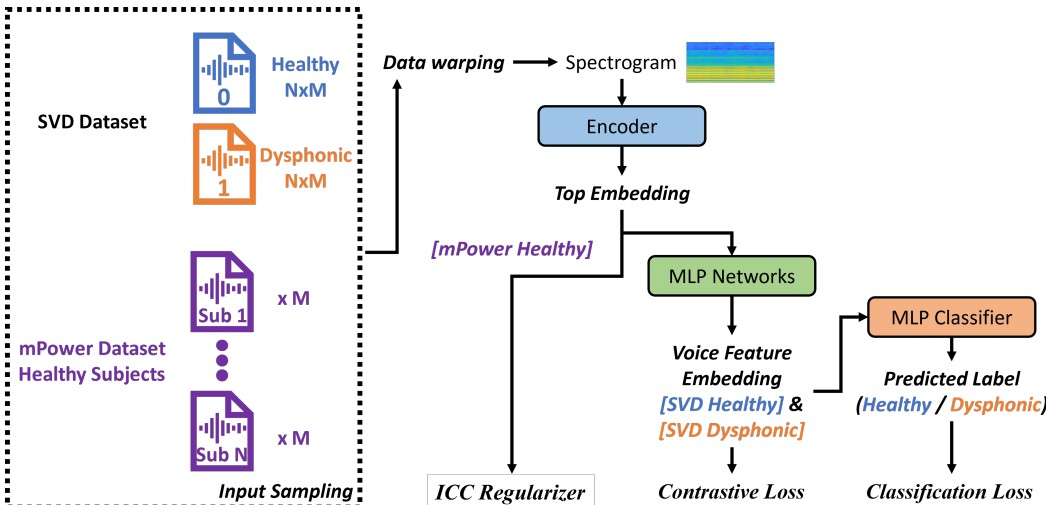

Figure C.4: The diagram of two-embedding output model structure. This is not used for any results in this paper.

**Convergence problems with the architecture from Figure C.3:** During experiments, we noticed that if the model has only one output, there is a conflict between the ICC regularizer for repeatability

and contrastive loss for dysphonic voice detection, i.e., the model has difficulties converging. We believe this problem is because the optimization objectives of the two components are so different. The ICC regularizer focuses on the repeatability within the subject; however, the contrastive loss focuses on the dysphonic sensitivity between the healthy and dysphonia groups. The inconsistent range of these two objectives requires careful fine-tuning of the ratio between the ICC regularizer and contrastive loss. We did not find a good loss ratio, so we abandon this structure.

**Convergence problems with the architecture from Figure C.4:** After the one-embedding output structure failed, we considered letting the ICC regularizer and contrastive loss optimize two different embeddings, and then we tested the two-embedding output structure (Figure C.4). The encoder's output is the top embeddings, and the MLP network is used to learn dysphonic voice feature embeddings. Then the ICC regularizer regularizes the repeatability of top embeddings, and the contrastive loss regularizes the dysphonic sensitivity of voice feature embeddings. The intuition is that since the dysphonic voice feature embeddings are a function of the top embeddings, the voice feature embeddings will also have the property of high repeatability. However, this structure failed to converge. The two-embedding output structure did not solve the conflict between the two loss's optimization objectives.

**Two-branch structure solves the convergence problems:** After the one- and two-embedding output structures failed, we designed the two-branch (three-embedding output) structure, solving the convergence problem. The top embeddings generated by the encoder are further converted to repeatable embeddings and dysphonic voice feature embeddings using two MLP networks. The ICC regularizer regularizes the repeatability of repeatable embeddings, and the contrastive loss regularizes the dysphonic sensitivity of voice feature embeddings. Due to the repeatable and dysphonic voice feature embeddings being independently transformed from the top embeddings, the top embeddings are endowed with both properties. These are the embeddings we used in the paper.

# D Summary of Used Databases in Paper

**VoxCeleb 1 [37]:** A large-scale speaker recognition dataset consisting of short video clips from YouTube. It includes over 100,000 utterances from more than 1,200 celebrities across various professions and demographics.

**VoxCeleb 2 [11]:** An extension of VoxCeleb 1, VoxCeleb 2 is an even larger dataset featuring approximately 1 million utterances from over 6,000 speakers. Together, VoxCeleb 1 and VoxCeleb 2 offer rich resources for training and evaluating speaker recognition models.

**VCTK (The Voice Cloning Toolkit) [52]:** VCTK is a speech dataset that includes recordings of various English accents. With over 44 hours of speech from 109 speakers, each speaking in their accent, VCTK provides a valuable resource for multi-accent speech synthesis and recognition research.

**MEEI (Massachusetts Eye and Ear Infirmary):** Full name is Kay Elemetrics Corp., Disordered Voice Database, Version 1.03 (CD-ROM), MEEI, Voice and Speech Lab, Boston, MA (October 1994). The MEEI Voice Disorders Database is a collection of speech samples from individuals with and without voice disorders. Participants are English speakers. It is often used in medical and clinical research to study voice pathology and develop systems to detect and analyze voice disorders. The MEEI database contains more than 1400 recordings of sustained phonations, which are collected from 53 healthy speakers and 657 speakers diagnosed with different types of dysphonia.

**SVD (Saarbrücken Voice Database) [57]:** The Saarbrücken Voice Database is a collection of voice recordings used for various phonetic and clinical studies. Participants are German speakers. It provides a comprehensive set of voice samples, including those from individuals with different voice disorders, aiding in the research of voice quality and characteristics. SVD database contains the voice recordings from more than 2000 speakers (428 healthy females, 259 healthy males, 727 dysphonic females, 629 dysphonic males).

**HUPA (Hospital Príncipe de Asturias) [1]:** Similar to MEEI and SVD, HUPA a collection of speech samples from individuals with and without voice disorders. Participants are Spanish speakers. HUPA contains /a/ sustained phonation recordings of 366 adult Spanish speakers (169 dysphonic and 197 healthy).

