# OpenReview forum: "Learning Repeatable Speech Embeddings Using An Intra-class Correlation Regularizer"
_NeurIPS.cc/2023/Conference — NeurIPS 2023 poster_

### Official Review · Reviewer_hWT5 · 2023-07-05

**Soundness:** 3 good
**Presentation:** 4 excellent
**Contribution:** 3 good
**Rating:** 7
**Confidence:** 2

**Summary:**

Starting with the observation that supervised embeddings should vary from one class to another but not be sensitive to variations within a given class, this paper proposes to add an intra-class correlation (ICC) regularization to the contrastive loss in representation learning. This new regularizer is derived from measurement theory and encourages the embeddings to have a low intra-class variance, whereas contrastive losses pushes more the embeddings towards high inter-class variances. After a simulation illustrating the "behaviour" of these losses with respect to inter- and intra-class variance, the authors report good improvements using the proposed ICC regularizer for three different tasks.

**Strengths:**

  - The paper is overall well written, the idea well explained and easy to follow and to understand
  - The ICC regularizer seems to really focus on intra-class variance, i.e. make all occurences of the same class have the same embeddings, where contrastive loss does both at the same time and never really focuses on minimizing the intra-class variance.
  - The method is relatively simple, very well explained and easy to understand. The simulation, comparison with contrastive loss and classification error rate is quite interesting.
  - When the embeddings are constrained, it is indeed useful to focus on minimizing the intra-class variance to separate classes. The explanation and results are convincing.

**Weaknesses:**

  - Missing comparison with other techniques mentioned in the related work (data augmentation, etc.). Is the proposed approach competitive with these methods? Could it be complementary? How does it compare to VICReg [4] for example
  - In the result section, the text mainly lists the results in the tables. Maybe a reformulation would allow more space for example to study the effect of the regularization weight

**Questions:**

Could that method apply to self-supervised approaches, where the notion of classes is less well defined?

Is it competitive with other methods mentioned at the beginning of the paper? Or could it be complementary?

Are the improvements with that method bigger on smaller datasets or are they independent of the dataset size? Is the method especially suited for some tasks and less for other (e.g. could that be useful for classification, or ASR?)?

minor remark:
  - sec 3.1: "represents the i-th samples" -> sample

**Limitations:**

yes

---

> ### Author Rebuttal · Authors · 2023-08-09
>
> We thank reviewer hWT5 for the comments.
>
> **Response to "Is ICC competitive with other techniques? Or could it be complementary?":** In Lines 30-31, we identify two primary groups of approaches for managing the intra-class variance: (1) enhancing the diversity of the training data, and (2) utilizing innovative learning algorithms. Below we discuss how the ICC is related to both types of approaches.
>
> 1. Our results show that the ICC regularizer is complementary to approaches belonging to the first group. For example, data augmentation was implemented in the speaker verification task (Section 4.1) and dysphonic voice detection task (Section 4.3) for both contrastive loss only and contrastive loss + ICC regularizer experiments. The ICC regularizer provides additional improvement on top of data augmentation. Implementing the ICC regularizer can indeed contribute to further minimizing intra-class variance, thus complementing those methods.
> 2. Our results show that the ICC regularizer provides benefit over GE2E, AngleProto, and SupCon (see Section 4.1). It is unclear that if ICC regularizer is complementary over all innovative learning algorithms. This will depend on the optimization criteria for each loss. For example, the algorithm proposed in (Rizve et al., 2021; [42] in the manuscript) focuses on using cosine similarity to construct the cost function. This is considerably different than the ICC and does not directly overlap with our ICC regularizer's functionality, we believe that our ICC regularizer could be a compatible addition to this algorithm. Since this focus does not directly overlap with our ICC regularizer's functionality, we believe that our ICC regularizer could be a compatible addition to this algorithm. In contrast, VICReg (Bardes et al., 2021; [4] in the manuscript) focuses on the variance of embedding vectors, which may be redundant with our regularizer.
>
> We will add a discussion about "Comparison between the ICC regularizer and other intra-class variance minimizing techniques" in Section 3.2 of revised manuscript.
>
> (Rizve et al., 2021) - Rizve, Mamshad Nayeem, et al. "Exploring complementary strengths of invariant and equivariant representations for few-shot learning." _Proceedings of the IEEE/CVF conference on computer vision and pattern recognition_. 2021.
>
> (Bardes et al., 2021) - Bardes, Adrien, Jean Ponce, and Yann LeCun. "Vicreg: Variance-invariance-covariance regularization for self-supervised learning." _arXiv preprint arXiv:2105.04906_ (2021).
>
> **Response to "more details about the effect of the regularization weight":** We will add more context to manuscript about regularization weight in revision based on the information provided Supplemental Material - Appendix Section 2.
>
> **Response to "apply to self-supervised approaches":** Any self-supervised method that can use contrastive-style training can also use the ICC. For example, consider the self-supervised deep learning methods proposed in (Falcon et al., 2020; Ciga et al., 2022): here, the anchor-positive pair could be treated as an intra-class data pair, and the anchor-negative pair treated as an inter-class data pair. This is an interesting avenue of future work, and we will mention this possibility in the revision at the end of the paper.
>
> (Falcon et al., 2020) - Falcon, W., & Cho, K. (2020). A framework for contrastive self-supervised learning and designing a new approach. arXiv preprint arXiv:2009.00104.
>
> (Ciga et al., 2022) - Ciga, O., Xu, T., & Martel, A. L. (2022). Self supervised contrastive learning for digital histopathology. Machine Learning with Applications, 7, 100198.
>
> **Response to "ICC regularizer effectiveness dependency on dataset size":** The improvements with the given method might be more pronounced on smaller datasets. Regularization techniques, in general, tend to be more useful for smaller datasets. However, it's hard to provide an unequivocal answer without a deeper analysis of the specific method and context. Depending on the method and the nature of the data, the impact might vary.
>
> **Response to "ICC regularizer suitability for specific tasks":** The method appears to be more suited for tasks where repeatability is a crucial factor, such as speaker verification and classification. It might be particularly beneficial for these applications, as consistency and reliability might be key in achieving high performance. On the other hand, for tasks like automatic speech recognition or pure end-to-end translation systems, where repeatability might be less critical, the method may not be as effective. Further analysis and experimentation would be required to determine the full scope of its applicability across different tasks and domains.

---

### Official Review · Reviewer_89AX · 2023-07-06

**Soundness:** 3 good
**Presentation:** 3 good
**Contribution:** 3 good
**Rating:** 5
**Confidence:** 2

**Summary:**

The paper introduces the ICC regularizer, a novel regularization technique that complements the contrastive loss to enhance the repeatability of embeddings. The authors illustrate the reason why the ICC regularizer better minimizes the intra-class variance than the contrastive loss, offering a fresh perspective on latent representation learning. The experimental results demonstrate that the ICC regularizer enhances the repeatability of learned embeddings. Consequently, embeddings with higher repeatability exhibit superior performance in downstream tasks.

**Strengths:**

1. This paper links the concept of repeatability, derived from measurement theory, to deep-learned embeddings, which is novel and interesting. The authors present an explanation of how the ICC regularizer effectively reduces intra-class variance compared to the contrastive loss, offering a novel perspective on latent representation learning.
2. To enhance the repeatability of the learned embeddings, this paper introduces a novel regularizer called the ICC regularizer, which is used as a regularizer for the contrastive loss.
3. The experimental results indicate that using the ICC regularizer leading to better performance in three downstream tasks .

**Weaknesses:**

My main concerns are mainly related to the experiments. Please refer to the following Questions section for details.

**Questions:**

1. In Section 4.2, ''GE2E Loss + ICC Regularizer'' shows superior speech naturalness on subjective evaluations. However, the objective metrics like speaker similarity scores and word error rate (WER) are missing.
2. In Section 4.2, the paper aims at zero-shot voice style conversion, which requires highly generalized speaker embedding. However, the authors train and evaluate the zero-shot voice style conversion on a small dataset called VCTK. I would like to ask if the generalization performance of the proposed method be further improved by using larger training datasets? Additional experiments on larger dataset would make the paper more convincing.
3. Although the audio samples with ICC regularizer is better, after listening to the audio samples in the supplementary materials, I think the audio quality should be further improved for practical scenarios. I would like to ask is this due to the used AutoVC model? Could the authors use the latest SOTA voice conversion model to improve the speech quality?

I would like to raise my scores if the authors have fully addressed my concerns.

**Limitations:**

Some limitations of the work are discussed. The potential negative societal impact of the work is not addressed.

---

> ### Author Rebuttal · Authors · 2023-08-09
>
> We thank reviewer 89AX for the comments.
>
> **Response to Question 1:** We evaluate two methods objectively by using a speaker similarity score and word error rate (WER). We use an opensource speaker encoder (huggingface.co/speechbrain/spkrec-ecapa-voxceleb) to calculate the speaker similarity score between target speaker's audio and transformed output. And we use Wav2Vec2 (huggingface.co/docs/transformers/model_doc/wav2vec2) to do ASR on the transformed output, and calculate the WER and character error rate (CER) by using *jiwer* module (pypi.org/project/jiwer/2.5.1/). The results are given as follows:
>
> (VC model used here is AutoVC.)
> ||Speaker Similarity Score|WER|CER|
> |--|--|--|--|
> |GE2E Loss|0.2231|0.5810|0.3817|
> |GE2E Loss + ICC Regularizer|0.2309|0.5109|0.3324|
>
> The results demonstrate that speaker embeddings with higher repeatability also result in better performance across all three metrics for voice style conversion.
>
> **Response to Question 2:** Using VCTK dataset solely for training is a widely adopted strategy in voice conversion researches, including in SOTA models (Tang et al., 2022; Casanova et al., 2022; Li et al., 2023; ). Other larger speech datasets such as LibriTTS and Common Voice are only used for testing in select papers (Li et al., 2023).
>
> Both LibriTTS and Common Voice are indeed substantial datasets, but they have different characteristics compared to VCTK.
>
> **VCTK**: Specifically designed for studying various speech processing tasks, including voice conversion. The dataset contains diverse accents but with relatively clean recordings.
>
> **LibriTTS**: LibriTTS is designed more for TTS (Text-To-Speech) tasks. It's derived from audiobooks, so the speaking style is primarily a reading style.
>
> **Common Voice**: This dataset has a broader goal of collecting voice data for a variety of languages to support voice technologies. The quality and style of recordings vary greatly in this dataset. What's more, many speakers have very limited recordings in this dataset.
>
> Voice conversion benefits from consistent and clean recordings, as it aims to capture and transfer speaker characteristics without extraneous noise or variability. And voice conversion requires capturing the nuances of individual speaker characteristics. So, compared to VCTK, the LibriTTS and Common Voice are not suitable for voice conversion model training.
>
> The above information gives some insights about why most VC research prefers VCTK over larger but noisier databases. However, as we found the reviewer's comment very interesting, we tried to train the AutoVC model on the Common Voice dataset and tested the model on the VCTK dataset under the unseen-to-unseen speaker scenario. The model is trained with "GE2E loss" (with and without the ICC) speaker encoder described in Section 4.1. We only report objective scores, including the speaker similarity score, WER and CER here, as there was not enough time to conduct AB preference experiments.
>
> (VC model used here is AutoVC.)
> |AutoVC|Speaker Similarity|WER|CER|
> |--|--|--|--|
> |GE2E Loss + train on VCTK|0.2231|0.5810|0.3817|
> |GE2E Loss + train on Common Voice|0.1934|0.5958|0.3942|
> |GE2E + ICC Regularizer + train on VCTK|0.2309|0.5109|0.3324|
> |GE2E + ICC Regularizer + train on Common Voice|0.2131|0.5308|0.3544|
>
> The results reveal that employing a larger dataset for training the VC model does not necessarily lead to improved performance. However, the proposed ICC regularizer is still effective when using a large training dataset and leads to a better performance on several metrics.
>
> (Tang et al., 2022) - Tang, Huaizhen, et al. "Avqvc: One-shot voice conversion by vector quantization with applying contrastive learning." _ICASSP 2022_. IEEE, 2022.
>
> (Casanova et al., 2022) - Casanova, Edresson, et al. "Yourtts: Towards zero-shot multi-speaker tts and zero-shot voice conversion for everyone." _ICML 2022_. PMLR, 2022.
>
> (Li et al., 2023) - Li et al. "Freevc: Towards High-Quality Text-Free One-Shot Voice Conversion." _ICASSP 2023_. IEEE, 2023.
>
> **Response to Question 3:** We use a SOTA voice conversion model to conduct the experiment outlined in Section 4.2. We selected a recently published work, FreeVC, for this purpose. Details of FreeVC can be found on GitHub at github.com/OlaWod/FreeVC and the paper is accessible at arxiv.org/abs/2210.15418.
>
> For the experiment, we utilized the version of FreeVC without speaker regularization  (w/o SR), as it has the fastest training speed compare other two versions and we have limited time for training this model. We trained it using the speaker encoder mentioned in our paper (Line 258-287). Both the training and testing were performed exclusively on the VCTK dataset. However, the training is longer than our expectation. We use the code provided by the author of FreeVC to train the model on an Nvidia A100 card. The model is still training, however we report an intermediate result below (the paper recommends 900k steps, results below are at ~500k steps) (Li et al., 2023). We report the objective scores, including the speaker similarity score, WER and CER for the 500k-steps-trained FreeVC model here.
>
> (VC model used here is FreeVC w/o SR, trained 500k steps.)
> ||Speaker Similarity Score|WER|CER|
> |--|--|--|--|
> |GE2E Loss|0.2753|0.2556|0.0755|
> |GE2E Loss + ICC Regularizer|0.2899|0.2163|0.0718|
>
> Even though we did not complete the FreeVC training, the results have already shown improved output performance over the results reported in the paper by using a SOTA voice conversion model. The reported speaker similarity scores, WER and CER, demonstrate that speaker embeddings with higher repeatability also result in better performance for voice style conversion tasks. It is important to note that the intermediate results reported herein cannot be directly compared to those in the paper as the model is still training.
>
> (Li et al., 2023) - Li et al. "Freevc: Towards High-Quality Text-Free One-Shot Voice Conversion." _ICASSP 2023_. IEEE, 2023.

---

### Official Review · Reviewer_Yotw · 2023-07-06

**Soundness:** 3 good
**Presentation:** 3 good
**Contribution:** 2 fair
**Rating:** 6
**Confidence:** 4

**Summary:**

The paper presents a simple but seemingly effective idea to regularize contrastive embedding extractors. The regularizer aims at the intra-class correlation to ensure the repeatability of the embedding.
The effectiveness is confirmed experimentally for the task of speaker verification, voice style conversion, and dysphonic voice detection.

**Strengths:**

The presented idea is well introduced and the illustration of the ICC regularizer action compared to the GE2E loss is insightful. The evaluation is sufficient.

**Weaknesses:**

The introduced regularizer uses 2nd order statistics only which could be a limiting factor on the embedding extractors but on the other hand this doesn't need to be required.

**Questions:**

The contrastive loss has a direct relationship to the mutual information, one could think that ICC is automatically included in the mutual information when only 2nd order statistics are taken into account, what could be the reason that this doesn't seem to be the case.

**Limitations:**

The introduced regularizer uses 2nd order statistics only.

---

> ### Author Rebuttal · Authors · 2023-08-09
>
> We thank reviewer Yotw for the comments.
>
> It is true that the ICC relies only on second-order statistics and it’s possible that it may not capture complex, nonlinear relationships within or between groups. That said, it does have several benefits over other more complex measures:
>
> 1. This simplicity often means fewer assumptions and parameters compared to some nonlinear measures. With small sample sizes, complex models that require estimation of numerous parameters may suffer from overfitting or instability, whereas the ICC may provide more robust estimates.
> 2. The ICC is a standard measure used in many fields, including psychometrics, medical research, and reliability engineering. Its widespread acceptance means that results based on the ICC can be easily compared with other studies or benchmarks.
> 3. The reliance on second-order statistics makes the ICC computationally efficient.
> 4. To the best of our knowledge there are no commonly-used measures of repeatability that are based on higher-order statistics.
>
> While the example contrastive loss in our manuscript, the GE2E loss shares some high-level goals with MI maximization (such as learning informative representations), as far as we know, it does not have a direct mathematical relationship to MI. The connection between contrastive learning and MI is more explicitly established in other contrastive methods like InfoNCE loss, which are specifically designed to approximate MI maximization. Those loss functions are less commonly used in speech, owing to the superior performance demonstrated by GE2E loss in speech tasks. While the Pearson correlation (different from the ICC measure we use here) does have a connection with the MI,  to the best of our knowledge, there is no standard or direct mathematical formula that connects ICC and MI. They are derived from different statistical principles.

---

### Official Review · Reviewer_efaw · 2023-07-07

**Soundness:** 3 good
**Presentation:** 3 good
**Contribution:** 3 good
**Rating:** 6
**Confidence:** 4

**Summary:**

In this paper, authors propose to use an intra-class correlation coefficient (ICC) regularizer as a complimentary component for contrastive loss to increase the repeatability of the supervised speech embedding. Authors conduct simulation for ICC regularizer and contrastive loss (GE2E loss) to explain their similarities and differences in the optimization criteria. Based on the experimental results on the speaker verification, voice style conversion, and a clinical application for detecting dysphonic three tasks, the authors claim that adding an ICC regularizer helps increase the repeatability and further improve the performance on downstream tasks.

**Strengths:**

1. This paper leverages the concept of repeatability and states that higher repeatability can contribute to improved performance in downstream tasks. Authors think about the importance of intra-class variance and the proposed ICC regularizer places greater emphasis on minimizing it to help increase repeatability. The simulation results in Section 3.2 explains the intuition clearly.

2. As a complementary component to contrastive loss, adding the proposed ICC regularizer successfully increase the repeatability and shows good performance in the three different tasks.

**Weaknesses:**

1. In Task 1, the performance of SV system is not good though adding the ICC regularizer improves its performance. In this paper, VGG-M-40 is trained on VoxCeleb1 & 2. The best result (3.78% EER) reported in Table 1 is not as good as other commonly used models for SV. In Chung et al. paper, the VGG-M-40 shows much worse performance than Thin-ResNet34 and Fast ResNet34. To make the experimental solid and results and claims more convincible, authors should pick a good baseline instead of use the VGG-M-40.

2. For better replicable, more experiment details should be provided, such as the training details of Task 1.

3. To have comprehensive evaluation results for Task 1, the evaluation metric minDCF can be used as well.

4. Visualization (such as t-SNE) can be added as a comparison to see if the extracted speech or speaker embeddings have smaller intra-class variance in the space, though their ICC is decreased.

**Questions:**

1. As this paper mentioned, the ICC regularizer requires hyperparameter fine-tuning. Are the hyperparameter tuned on a validation set or a development set?

2. As this paper mentioned in Section 3.2, though the ICC increased, do the learned embeddings are more compacted within-class and more separate between-class in the overall embedding space? Authors can use t-SNE to visualize them.

**Limitations:**

The authors do not analyze the potential negative societal impact.

---

> ### Author Rebuttal · Authors · 2023-08-09
>
> We thank reviewer efaw for the comments.
>
> **Response to Weakness 1:** In this study, our focus is not on demonstrating that the proposed method can reach SOTA performance across various tasks. Rather, we aim to illustrate that the proposed ICC regularizer can improve the repeatability of learned embeddings, improving the performance of downstream applications. Additional experiments demonstrate that even with models better than the ones used in the paper, the proposed ICC regularizer can still improve performance. For example, we re-run VC task with a SOTA VC model named FreeVC with our speaker encoder (manuscript Line 258-287) to conduct the experiment outlined in Section 4.2. (For details please see response to Reviewer 89AX.)
>
> (VC model used here is FreeVC w/o SR.)
> ||Speaker Similarity Score|WER|CER|
> |--|--|--|--|
> |GE2E Loss|0.2753|0.2556|0.0755|
> |GE2E Loss + ICC Regularizer|0.2899|0.2163|0.0718|
>
> While this model is still training (authors suggest 900k steps), the results above represent an intermedia evaluation of the model. The higher speaker simialrity score, the better performance of model. The lower WER & CER, the better performance of model. As the results show, even at this intermediate evaluation, the consistent pattern from the SOTA VC model serves as tangible evidence that including the ICC regularizer with existing cost functions demonstrates consistent improvements across several performance metrics.
>
> We ran out of time for training the Fast ResNet34 model. When the FreeVC model training is complete, we will train the Fast ResNet34 model referenced in Chung et al.'s paper for the speaker verification task and refine and optimize our methodology.
>
> **Response to Weakness 2:** In the revision, we will add following contents to Section 4.1 Experiment Setup:
>
> *During training, we employ the Adam optimizer, maintaining a static learning rate of 0.001 without implementing any learning rate schedule. The dropout rate is set to 0.2 for all dropout layers. As for data augmentation: (1) we use variation in input audio length by randomly fixing the audio duration within a range of 1.5 to 3.0 seconds, and (2) we add Gaussian noise with an SNR randomly selected between 15 to 60 dB. No other augmentation methods are used.*
>
> **Response to Weakness 3:** We follow the reviewer's comment and add the minDCF metric to Table 1. For the minDCF, hyper-parameters of $P_{target}=0.05$, $C_{fa}=C_{miss}=1$.
>
> ||minDCF|
> |--|--|
> |GE2E|0.2925|
> |GE2E+ICC|0.2778|
> |AngleProto|0.2809|
> |AngleProto+ICC|0.2790|
> |SuperCon|0.2791|
> |SuperCon+ICC|0.2597|
>
> The results demonstrate that the ICC regularizer improves model performance relative to the minDCF metric on the TI-SV task. The improvement in the metric is clear for all three baseline models to which the ICC regularizer is added.
>
> **Response to Weakness 4 and Question 2:** We follow the reviewer's comment and generate the figure of speaker embeddings t-SNE projection. We use two trained speaker encoders from Section 4.1, i.e., GE2E and GE2E+ICC, to generate the speaker embeddings. For the audio data, we randomly select 12 speakers from the test dataset and randomly select 10 samples from each speaker. After using the two encoders to extract all embeddings for all samples, we run t-SNE on these embeddings, and the results are shown in Figure B (in global response PDF).   Please take a look at Speaker No. 4 & 7: in Figure B(a), the two speakers’ embeddings are mixed with other speakers, but in Figure B(b), the two speakers become separate from other speakers due to the fact that the ICC regularizer improves the repeatability of embeddings relative to speaker identities.
>
> **Response to Question 1:** The hyper-parameter is tuned on the development dataset. The EER for subjects in the development dataset are used to determine the optimized hyperparamter. We will be sure to highlight this in the revision.

---

> > ### Comment · Reviewer_efaw · 2023-08-20
> >
> > I'd like to thank the authors for their feedback and some new experimental results. The explanations almost solve all my concerns. Although authors explained and mentioned that “our focus is not on demonstrating that the proposed method can reach SOTA performance across various tasks”, I totally understand that. Authors do not have to use the SOTA SV baseline or achieve the SOTA performance. But the performance of VGG-M-40 reported in this paper is much worse than current common SV system performance (e.g., ResNet and ECAPA-TDNN with around 1.0% EER on VoxCeleb1-O). I am looking forward to seeing the results of using a better SV baseline system such as Thin ResNet34 or Fast ResNet34.
> >
> > I will reconsider the review in light of the provided clarifications and raise the score.

---

### Official Review · Reviewer_wtbk · 2023-07-07

**Soundness:** 3 good
**Presentation:** 3 good
**Contribution:** 3 good
**Rating:** 6
**Confidence:** 4

**Summary:**

In this paper, the concept of repeatability from measurement theory was introduced for representation learning. Intraclass Correlation (ICC) was proposed as an evaluation metric and the ICC regularizer was used as an additional element to contrastive loss for training. This aims to enhance the repeatability of deep-learned embeddings.
Some synthetic examples and intuitive reasoning were used to illustrate why the ICC regularizer is superior in minimizing intra-class variance compared to contrastive loss. The ICC regularizer was assessed across three speech tasks: speaker embeddings for Text-Independent Speaker Verification (TI-SV) and zero-shot voice style conversion, along with voice feature embeddings for a clinical application. The experiments indicate that the ICC regularizer can boost the repeatability of learned embeddings and perform better in downstream tasks.

**Strengths:**

The paper proposed a novel metric of ICC based on measurement repeatability. Although the term may be a little confusing, the motivation
and intuition were well explained. The experimental results on three speaker/speech tasks are convincing.

**Weaknesses:**

To show the repeatability of speaker embedding for TI-SV task, more controlled experiments may be necessary to prove the effectiveness of ICC loss in addition to the ICC scores: for example, adding noise to the testing sample, or switching languages of the same speaker as it was tested in the NIST Speaker Recognition Evaluation.

For results on dysphonic voice detection in Table 3, other baseline models were definitely over-trained compared to the proposed model.
Why do you train your own models instead of quoting the numbers from the original papers? (see line 334-335). The results in Table 3 may be unfair comparisons.



**Questions:**

See the comments above in the weakness section.

**Limitations:**

No discussion on negative societal impact was included in the paper

---

> ### Author Rebuttal · Authors · 2023-08-09
>
> We thank reviewer wtbk for the comments.
>
> **Response to "more controlled experiments may be needed":** We follow the reviewer's suggestion and run more testing on audio samples with noise added.    We evaluate three SNR levels, 30, 35 and 40 dB. For each input audio, we randomly generated Gaussian noise and scale the noise to the desired energy level. We add the audio and the noise to get the noisy audio at the desired SNR level. The EER results are provided in the following table. The results show that adding the ICC regularizers improves the original loss for all 3 SNR levels evaluated. Furthermore, the performance gain is greater for the lower SNR (30dB) than for the other two conditions, although the differences are small.
>
> |Method|SNR=40dB|SNR=35dB|SNR=30dB|
> |--|--|--|--|
> |GE2E|5.64%|7.00%|9.04%|
> |GE2E+ICC|5.34%|6.76%|8.66%|
> |AngleProto|4.77%|5.76%|7.66%|
> |AngleProto+ICC|4.52%|5.59%|7.34%|
> |SupCon|4.35%|5.21%|6.25%|
> |SupCon+ICC|4.21%|5.09%|6.10%|
>
> **Response to "why not quoting the numbers from the original papers"**: We can quote the reported numbers from the original paper for Zhang, Harar, and Huckvale, although the results from Verde are not comparable. Firstly, it's important to realize that (Verde et al., 2018), did not utilize the complete SVD dataset; the authors relied on a subset of well-separated samples, painting an overoptimistic performance of how well the method works. Authors of the three baseline methods (Harar et al., 2017; Huckvale et al., 2021; Zhang et al., 2022) all express this concern about the (Verde et al, 2018) paper.
>
> We list the methods' SVD testing accuracy of the original papers and our rebuilt results as follows:
>
> |Methods|SVD Testing Acc - Original Paper|SVD Testing Acc - Our Rebuilt|
> |--|--|--|
> |Our Proposed Method|N/A|0.7289|
> |Zhang et al. (2022)|0.7077|0.7077|
> |Harar et al. (2017)|0.6808|0.6914|
> |Huckvale et al. (2021)|0.6974|0.6255|
>
> We will add this information to Table 3 in the revision. The new data does not impact our conclusion in the manuscript as our method outperforms the three baseline methods, Harar et al. (2017), Huckvale et al. (2021), and Zhang et al. (2022). And the following additional context explains why we had to rebuild the models:
>
> Two baseline methods (Verde et al., 2018 and Huckvale et al., 2021) both employ SVM-based algorithms. Yet, neither of these studies provides details on the training accuracy within their papers. To ensure a comprehensive and fair evaluation, we decided to report the training accuracies for our method as well as the baseline methods. This required us to reproduce their methods, thereby allowing for a more transparent and reliable comparison.
>
> Except for Zhang et al. (2022), all other baseline methods do not provide enough information to reproduce their work, which may lead to degraded performance of our rebuilt model compared to their original works. For example, Hara et al. did not release the code publicly, and Huckvale et al. did not provide their feature selection criteria, how the features were selected, or what the final list of features used in the model was.
>
> (Harar et al, 2017) - Harar, Pavol, et al. "Voice pathology detection using deep learning: a preliminary study." _2017 international conference and workshop on bioinspired intelligence (IWOBI)_. IEEE, 2017.
>
> (Verde et al., 2018) - Verde, Laura, Giuseppe De Pietro, and Giovanna Sannino. "Voice disorder identification by using machine learning techniques." _IEEE access_ 6 (2018): 16246-16255.
>
> (Huckvale et al., 2021) - Huckvale, Mark, and Catinca Buciuleac. "Automated detection of voice disorder in the Saarbrücken voice database: Effects of pathology subset and audio materials." _Proceedings of the Annual Conference of the International Speech Communication Association, INTERSPEECH_. Vol. 6. International Speech Communication Association (ISCA), 2021.

---

> > ### Comment · Reviewer_wtbk · 2023-08-21
> >
> > Thanks for more experimental results and explanations on baselines. However, I keep the score unchanged due to no additional value added to the main theme of the paper. It would be better to choose the right baselines in the future due to the various reasons mentioned in the rebuttal.

---

### Official Review · Reviewer_Qvpp · 2023-07-26

**Soundness:** 2 fair
**Presentation:** 3 good
**Contribution:** 2 fair
**Rating:** 5
**Confidence:** 4

**Summary:**

This paper proposes to use a traditional intra-class correlation coefficient (ICC) to assess the repeatability of speech embeddings learnt by neural network. The proposed ICC regularizer has characteristics similar to well-known contrastive loss which aims to minimize the intra-class variance and maximize the inter-class variance of training data. The authors propose to use the ICC regularizer not in isolation but in combination with the contrastive loss. This paper performs several types of experiments based on speech processing tasks such as speaker verification, voice style conversion, and dysphonic voice detection to demonstrates the effectiveness of the proposed regularizer. These tasks focus on the importance of detecting the repeatability for the target data.

**Strengths:**

- The considered scenario is interesting and important to the community. The proposed method is simple yet effective in assessing the repeatability in the target data.
- This paper assesses the effectiveness of the proposed method not only with a single speech task but also with several tasks based on voice quality evaluation. The improvements look sufficient.

**Weaknesses:**

- ICC is very traditional metric, which is widely used for various data classification algorithms.
- The proposed ICC regularizer shouldn't be used in isolation.
- Lack of some experiments and details of experimental setup (see questions below).

**Questions:**

- It looks like that the proposed ICC regularizer corresponds to the special case of contrastive loss focused only on positive sample pairs. Although there are many variants in contrastive loss, the basic concept is to increase a capability of category classification of data by leveraging both positive and negative pairs. I am wondering if the training only with conventional contrastive loss could provide some improvements similar to the proposed method, for example, by simply changing the ratio of positive sample pairs (reducing negative sample pairs) in the last half of the training stage. What are advantages that cannot be obtained by just changing the ratio of positive and negative pair data?
- It is difficult to understand the effectiveness of the proposed method correctly because experimental setups including data are very briefly written. For example, in the experiment described in Section 4.1, what are the training strategy of the model such as an optimizer, a learning rate scheduling, other regularization techniques (specAugment, DropConnect, etc.), data augmentation, and so on?
- In addition, a summary of data used for each experiment (a few more details) would be very much helpful.

**Limitations:**

The same comments as one of weaknesses. The proposed ICC regularizer shouldn't be used in isolation but be used in combination with a conventional contrastive loss as clearly mentioned in the paper.

---

> ### Author Rebuttal · Authors · 2023-08-09
>
> We thank reviewer Qvpp for the comments.
>
> **Response to "ICC is a traditional metric and widely used for various classification tasks":** The ICC is primarily a statistical measure that describes how strongly units in the same group resemble each other. It's typically used in various research fields like psychology, sociology, or medicine, where repeated measurements are taken on the same subjects to assess the reliability of ratings or measurements. The ICC is not a commonly used metric in machine learning classification tasks which typically rely on accuracy, F1, AUC-ROC, etc. Sometimes the ICC is used in machine learning classification tasks related to clinical applications (Ugga et al. 2021) for assessing the reliability of models but we are not aware of other classification papers which utilize ICC.
>
> Based on existing literature, employing the ICC as a measurement tool for assessing the quality of learned embeddings represents a novel contribution. Utilizing the ICC regularizer to ensure the model learns consistent and repeatable latent representations further distinguishes our approach as innovative. Within representation learning, our introduction of the ICC metric and ICC regularizer constitutes a significant contribution.
>
> (Ugga et al. 2021) - Ugga et al. "Meningioma MRI radiomics and machine learning: Systematic review, quality score assessment, and meta-analysis." _Neuroradiology_ 63 (2021): 1293-1304.
>
> **Response to "ICC regularizer shouldn't be used in isolation":** As outlined in Lines 224-225 of the manuscript, we advocate for using the ICC as a regularizer rather than as a stand-alone loss function. The ICC regularizer is designed to act as a complementary component in learning repeatable latent representations. It's important to recognize that most regularizers do not function as stand-alone loss functions; instead, the ICC regularizer is intended to work synergistically with other components of the loss function to enhance the reliability and coherence of the learned embeddings. As additional evidence of the utility of the regularizer, we have conducted the following additional experiments:
>
> (1) Run speaker verification (SV) testing on audio samples with noise added. We evaluate three SNR levels, 30, 35 and 40 dB. Adding the ICC regularizers improves the original loss for all 3 SNR levels evaluated. (Details please see response to Reviewer wtbk.)
>
> |Method|SNR=40dB|SNR=35dB|SNR=30dB|
> |--|--|--|--|
> |GE2E|5.64%|7.00%|9.04%|
> |GE2E+ICC|5.34%|6.76%|8.66%|
> |AngleProto|4.77%|5.76%|7.66%|
> |AngleProto+ICC|4.52%|5.59%|7.34%|
> |SupCon|4.35%|5.21%|6.25%|
> |SupCon+ICC|4.21%|5.09%|6.10%|
>
> (2) Evaluated the two methods in Section 4.2, voice conversion (VC), objectively on three different metrics: speaker similarity score, word error rate (WER), and character error rate (CER). (Details please see response to Reviewer 89AX.)
>
> (VC model used here is AutoVC)
> ||Speaker Similarity Score|WER|CER|
> |--|--|--|--|
> |GE2E Loss|0.2231|0.5810|0.3817|
> |GE2E Loss + ICC Regularizer|0.2309|0.5109|0.3324|
>
> The higher speaker similarity score, the better performance of model. The lower WER & CER, the better performance of model.
>
> (3) Re-run VC task with a SOTA VC model named FreeVC with our speaker encoder (manuscript Line 258-287) to conduct the experiment outlined in Section 4.2. (For details please see response to Reviewer 89AX.)
>
> (VC model used here is FreeVC w/o SR.)
> ||Speaker Similarity Score|WER|CER|
> |--|--|--|--|
> |GE2E Loss|0.2753|0.2556|0.0755|
> |GE2E Loss + ICC Regularizer|0.2899|0.2163|0.0718|
>
> While this model is still training, the results from an intermedia evaluation of the model tell a consistent story: when the regularizer is added to existing cost functions, performance improves on several metrics.
>
> **Response to "ICC regularizer corresponds to the special case of contrastive loss focused only on positive sample pairs":** The ICC regularizer cannot be posed as a special case of contrastive loss focused only on positive sample pairs as it requires both positive and negative sample pairs. The ICC regularizer and contrastive loss have similarities in their optimization criteria: both aim to minimize the intra-class variance by using positive sample pairs and maximize the inter-class variance using negative sample pairs. However, compared to contrastive loss, the ICC regularizer results in a better trade-off between minimizing intra-class variance and maximizing inter-class variance when inter-class variance is relatively larger than intra-class variance. This was shown in Fig. 1 in the manuscript.
>
> **Response to "changing the ratio of positive sample pairs (reducing negative sample pairs)":** Reducing negative sample pairs could help the loss focus on minimizing the intra-class variance; however this comes at the cost of the inter-class variance decreasing due to the imbalanced training strategy. We use our GE2E loss simulation to demonstrate this: all configurations and parameters are kept the same as those from the simulation in Fig. 1 in the manuscript, except we add a scalar to the negative pairs score in the GE2E loss to simulate the impact of reducing negative sample pairs during training. We set the scalar to 0.25, i.e., the ratio of negative sample pairs to positive sample pairs is 1:4. The results of our experiment are shown in Figure A (available in the global response PDF). The figure demonstrates that the trajectory of the optimal solution is in the direction of decreased inter-class variance. This is clearly in an undesired direction as a good regularizer promotes lower intra-class variance, but higher inter-class variance. Comparing this solution to the one from the ICC in Fig. 1 in the manuscript, we clearly see that the ICC does both; however simply changing the ratio of samples between the two classes only reduces the intra-class variance.
>
> **Response to "experiments and details of experimental setup":** We will add these in the revision. Please see the **Global Response**.

---

> > ### Comment · Reviewer_Qvpp · 2023-08-16
> > **Lowering the ratio of negative sample pairs**
> >
> > Thank you for the response and some new results. I understood that on the Monte Carlo simulation lowering the ratio of negative sample pairs (adding the negative pairs score) on GE2E loss provides undesirable impact for the problem setting in this paper. I assume that you have done the simulation with the same ratio between positive and negative sample pairs during an entire stage of training (in other words, all the epochs). However, the appropriate ratio of positive and negative sample pairs for each downstream task should be different. It can be estimated with a development set. In that sense, it is still unclear and my concern if the proposed method yields sufficient improvements over such baseline systems in actual downstream tasks.

---

> > > ### Author Response · Authors · 2023-08-17
> > >
> > > We thank the Reviewer Qvpp for engaging with our rebuttal and posing a new concern.
> > >
> > > **In Reviewer Qvpp's initial review,** the reviewer was concerned that "_the proposed ICC regularizer corresponds to the special case of contrastive loss focused only on positive sample pairs._" Our rebuttal clarified that this was not the case as the proposed regularizer requires both positive and negative pairs. It is impossible to implement using only positive samples. The reviewer further questioned "_What are the advantages that cannot be obtained by just changing the ratio of positive and negative pair data?_" We showed via simulation that the loss surface achieved by intentional class imbalance can result in optimal solutions in the direction of decreased inter-class variance; clearly an undesirable outcome.
> > >
> > > **In Reviewer Qvpp's most recent review,** the reviewer suggests that we should have optimized the imbalance ratio in our baseline results via a development set: "_However, the appropriate ratio of positive and negative sample pairs for each downstream task should be different. It can be estimated with a development set._"
> > >
> > > In a literature review, we found no evidence to suggest that using an imbalanced ratio of positive/negative pairs can enhance model performance over a balanced baseline. On the contrary, much of the research (e.g., Wang et al., 2021; Dorigatti et al., 2022; Vito et al., 2022) concentrates on methods to intentionally balance this ratio, particularly in cases of imbalanced datasets.
> > >
> > > Given:
> > > - our empirical results we provided last time that demonstrate the negative impact of class imbalance in contrastive loss;
> > > - no found evidence in the literature that deliberate class imbalance in contrastive loss improves performance;
> > > - our demonstration last time that the proposed regularizer is different from the special case of contrastive loss focused only positive pairs,
> > >
> > > we do not see any reason to believe that an asymmetric contrastive loss should be considered as a baseline over the standard balanced approach. We have not come across any papers that use an asymmetric contrastive loss as a baseline. Furthermore, we have not come across any papers that use cross-validation to set the imbalance ratio for optimizing performance.
> > >
> > > (Wang et al., 2021) - Wang, Peng, et al. “Contrastive learning based hybrid networks for long-tailed image classification.” _Proceedings of the IEEE/CVF conference on computer vision and pattern recognition_. 2021.
> > >
> > > (Dorigatti et al., 2022) - Dorigatti, Emilio, et al. “Robust and Efficient Imbalanced Positive-Unlabeled Learning with Self-supervision.” _arXiv preprint arXiv:2209.02459_ (2022).
> > >
> > > (Vito et al., 2022) - Vito, Valentino, and Lim Yohanes Stefanus. “An Asymmetric Contrastive Loss for Handling Imbalanced Datasets.” _Entropy_ 24.9 (2022): 1303.

---

> > > > ### Comment · Reviewer_Qvpp · 2023-08-21
> > > >
> > > > Thank you again for sharing some papers regarding the adjustment of positive and negative sample pairs. I also checked several papers and found no clear evidence to suggest that using the imbalanced ratio of positive/negative pairs provides an improvement although there were some papers just mentioning that they used the imbalanced ratio of pairs without any comparisons regarding the ratio. Based on these information provided by the authors, I will raise the score.

---

### Author Rebuttal · Authors · 2023-08-09

We thank all reviewers for the comments.

**Response to "discussion on potential negative societal impact":**

Methods for learning new feature representations that focus on separability between classes can amplify biases that exist in the data. This is a well-known problem and it can occur when the data used to train the representation model is biased. This is especially problematic in high-stakes applications like healthcare, where biased predictions or decisions can lead to unequal treatment or access. Safe deployment of models based on the feature representations proposed herein will require thorough validation to detect potential biases and mitigation strategies for dealing with them. We will add this discussion to the revised paper.

**Response to "experiments and details of experimental setup":**

In the revision, we will add the following information to Section 4.1 Experiment Setup:

*During training, we use the Adam optimizer, maintaining a static learning rate of 0.001 without implementing any learning rate schedule. The dropout rate is set to 0.2 for all dropout layers. As for data augmentation: (1) we use variation in input audio length by randomly fixing the audio duration within a range of 1.5 to 3.0 seconds, and (2) we add Gaussian noise with an SNR randomly selected between 15 to 60 dB. No other augmentation methods are used.*

If accepted, we will add the following information about datasets we used in the supplemental material:

**VoxCeleb 1**: A large-scale speaker recognition dataset consisting of short video clips from YouTube. It includes over 100,000 utterances from more than 1,200 celebrities across various professions and demographics.

**VoxCeleb 2**: An extension of VoxCeleb 1, VoxCeleb 2 is an even larger dataset featuring approximately 1 million utterances from over 6,000 speakers. Together, VoxCeleb 1 and VoxCeleb 2 offer rich resources for training and evaluating speaker recognition models.

**VCTK** (The Voice Cloning Toolkit): VCTK is a speech dataset that includes recordings of various English accents. With over 44 hours of speech from 109 speakers, each speaking in their accent, VCTK provides a valuable resource for multi-accent speech synthesis and recognition research.

**MEEI** (Massachusetts Eye and Ear Infirmary): The MEEI Voice Disorders Database is a collection of speech samples from individuals with and without voice disorders. Participants are English speakers. It is often used in medical and clinical research to study voice pathology and develop systems to detect and analyze voice disorders. The MEEI database contains more than 1400 recordings of sustained phonations, which are collected from 53 healthy speakers and 657 speakers diagnosed with different types of dysphonia.

**SVD** (Saarbrücken Voice Database): The Saarbrücken Voice Database is a collection of voice recordings used for various phonetic and clinical studies. Participants are German speakers.  It provides a comprehensive set of voice samples, including those from individuals with different voice disorders, aiding in the research of voice quality and characteristics. SVD database contains the voice recordings from more than 2000 speakers (428 healthy females, 259 healthy males, 727 dysphonic females, 629 dysphonic males).

**HUPA** (Hospital Príncipe de Asturias): Similar to MEEI and SVD, HUPA a collection of speech samples from individuals with and without voice disorders. Participants are Spanish speakers. HUPA contains /a/ sustained phonation recordings of 366 adult Spanish speakers (169 dysphonic and 197 healthy).

---

### Comment · Area_Chair_MiTb · 2023-08-15

Dear reviewers,

The authors have uploaded their rebuttal.  Please take time to go over it.  If you have any further questions or concerns regarding the authors' rebuttal, please start a discussion.   If you are willing to adjust your scores after reading the rebuttal, please do.

Thanks,

AC

---

### Decision · Program_Chairs · 2023-09-21

**Decision:**

Accept (poster)

**Comment:**

This paper introduces an intra-class correlation (ICC) regularizer to promote repeatability of speech embeddings and thus improve their invariance in the downstream tasks.  The proposed ICC regularizer works as a complementary to the commonly used contrastive loss.  It can be shown to give rise to a better trade-off between minimizing intra-class variance and maximizing inter-class variance when inter-class variance is relatively larger than intra-class variance.  Experiments are carried out on multiple tasks on a variety of downstream tasks such as speaker verification, zero-shot voice style conversion and dysphonic voice detection.  Globally the idea is interesting and the work is solid.  Most of the concerns on the methodology and experiments have been cleared in the rebuttal.  The reviewers have reached a consensus to accept.